# Reliability of Tooth Width Measurements Delivered by the Clin-Check Pro 6.0 Software on Digital Casts: A Cross-Sectional Study

**DOI:** 10.3390/ijerph19063581

**Published:** 2022-03-17

**Authors:** Milagros Adobes Martin, Erica Lipani, Laura Bernes Martinez, Alfonso Alvarado Lorenzo, Riccardo Aiuto, Daniele Garcovich

**Affiliations:** 1Department of Dentistry, Universidad Europea de Valencia, 46010 Valencia, Spain; erica.lipani@outlook.it (E.L.); laura.bernes@universidadeuropea.es (L.B.M.); daniele.garcovich@universidadeuropea.es (D.G.); 2Department of Oral Surgery, Universidad de Salamanca, 37007 Salamanca, Spain; alfonsoalvaradolorenzo@gmail.com; 3Department of Biomedical, Surgical, and Dental Science, University of Milan, 20122 Milan, Italy; riccardo.aiuto@unimi.it

**Keywords:** orthodontic model analysis, measuring agreement, digital models, 3D diagnosis and treatment planning

## Abstract

The purpose of this study was to assess the accuracy and reliability of tooth width measurements and Bolton Indices delivered by the Clin-Check Pro 6.0 software (Align Technology, San José, CA, USA). Fifty-four plaster casts were selected and measured with a manual calliper by a trained and calibrated observer. The data gathered were compared with those delivered by the software on the corresponding fifty-four virtual casts. The method reliability of the software was assessed by comparing the measurements performed by the software on 201 pairs of clin-checks corresponding to two consecutive treatment phases. Accuracy and reliability were statistically assessed using a mixed model. The software tends to provide larger widths compared with the manual method. Statistically significant differences were found in 23 teeth. At a global level, the mean difference between the methods was −0.19 mm, with a Dahlberg error of 0.24 mm and an intraclass correlation coefficient of 0.98. The Bolton Indices delivered by the two methods had a moderate correlation (ICC = 0.59; 0.69). The within method reliability of the software was extremely high. Tooth width measurements delivered by the software, despite the positive bias, can be considered accurate and clinically acceptable for all teeth except molars. The Bolton Indices made available by the software are not accurate and clinically acceptable, especially in the case of mandibular excess.

## 1. Introduction

Tooth width measurements are of great interest in dentistry, anthropology, and legal medicine [1]. A precise assessment of tooth width is a key point of orthodontic diagnosis, being pivotal for space analysis and to estimate tooth size discrepancies. An excellent occlusion depends, among other factors, on a harmonious intermaxillary tooth-size relationship [2]. This kind of assessment is of particular importance, being the reported prevalence of tooth size discrepancies of about 20 to 30% in both the general and orthodontic population [3,4].

Despite being proposed in 1952 and based only on 55 patients with normal occlusion, mainly orthodontically treated cases, the classic Bolton tooth size analysis is still the gold standard for diagnosing tooth size discrepancies between the upper and lower jaw [5].

Tooth width can be classically assessed by a manual calliper on conventional plaster casts [6]. In the actual orthodontic scenario, there is a clear trend towards a digital workflow relying on digital casts obtained by 3D optical scanning devices. Previous studies generally agree that the digital work-flow itself can be trusted for clinical purposes, and that digital casts seem to be a reliable alternative to traditional plaster casts for the routine measurements performed in orthodontics [7]. Due to the fast development of digital technologies, users are continually offered a wide variety of hardware and software solutions and therefore it is important to test their accuracy and reliability before using their finding for decision-making in a clinical setting [8]. 

Most of the current aligner systems offer orthodontic software programs, providing digital platforms to visualize, through a virtual set-up, the occlusal results of the treatment as well as the magnitude of the planned tooth movement. These platforms provide access to data such as tooth and arch width measurements, Bolton ratios, and offer a wide range of diagnostic tools. The number of clear aligners patients and the types of malocclusions treated with aligners are on the rise. Invisalign (Align Technology, San José, CA, USA) is currently the most prevalent clear aligner treatment method with over 8 million Invisalign treated cases worldwide [9]. Invisalign Clin-Check Pro 6.0 software (Align Technology, San José, CA, USA) makes a great amount of data available to clinicians and researchers in a direct and immediate way, but are these data reliable? The objectives of our study were to evaluate the agreement between the tooth width measurements made available by the software and those gathered manually on plaster casts, to assess the agreement of the tooth width measurements performed by the software on the same tooth at different time points (within method agreement) and to evaluate how the accuracy of the system can be affected by tooth misalignment.

## 2. Materials and Methods

Digital casts were selected from the archives of the Department of Orthodontics of the Universidad Europea de Valencia, Spain, and two private practices in the Valencia area. The design of this cross-sectional and observational study was approved by the European University of Valencia Ethics Committee on Human Research (Spain; reference number: CIPI/20/233).

All digital scans were obtained through an ITero Element scanning device (Align Technology, Orchard Parkway, San José, CA, USA). The power analysis conducted to set the sample size estimated that 48 dental casts would provide more than 95% power to detect significant differences with an effect size of 0.6 at an a = 0.05 level of significance as reported also by previous studies [10].

A first pool of 54 plaster casts and the corresponding digital casts were selected according to the following inclusion criteria and downloaded and stored in STereoLithography (STL) format: dental scans and plaster casts of excellent quality, free of deformation; all permanent teeth erupted from right to the left first molar in both jaws; non previous history of orthodontic treatment; no anomalies of number, form or structure; and no carious lesions, fractures, or restorations that can alter the mesiodistal dimension [11].

Plaster casts were obtained from upper and lower impressions taken in the same session as the intra-oral scan using a two-step putty and wash impression with polyvinyl siloxane (PVS) impression material (Aquasil Ultra; Dentsply Caulk, Milford, CT, USA) with a plastic dental tray, following the manufacturer’s protocol. The impressions were then poured with Zeta Orthodontic Stone plaster (Zhermack SpA, Badia Polesine, Italy) shortly after impression taking. 

The mesiodistal width of every tooth from the first right molar to the first left molar in each jaw was recorded by measuring the greatest distance between physiological contact points on the proximal surfaces. Measurements were performed using a fine tip digital calliper (Masel Orthodontics, Carlsbad, CA, USA), with an accuracy of 0.01 mm. The calliper was orientated parallel to the occlusal surface and perpendicular to the long axis of the tooth. Each tooth was measured twice, and the mean value was registered. According to what was reported by Flores-Mir et al., 2003, if the second measure differed more than 0.2 mm from the first measure, then the tooth was measured again and the mean value of the three measurements was registered [10]. All measurements were taken by a trained operator (E.L.). The operator was trained by one senior team member (D.G.) through theoretical and practical sessions in which the use of the calliper and the landmark positioning and location were explained and performed, every training session involved the repeated measurement of three study casts, not belonging to the study group. The training session ended when the operator was able to obtain a main difference between repeated measurements, less or equal than 0.2 mm. The casts belonging to the study group were measured over 11 sessions. A maximum of three to six pairs of digital casts were assessed per session to reduce the error derived from ocular fatigue. To assess the intra-operator reliability, a group of 10 randomly selected casts, belonging to the study group, was measured twice at a two-week interval [4]. To assess the inter-operator reliability, a second operator (D.G.) measured the same dental cast. The reliability of the second operator was assessed in a previous odontometric study [1], the related reliability data are provided in Appendix A. The mesio-distal widths were registered on an Excel datasheet (Microsoft Office for Mac 2011 package format). The anterior and overall Bolton Indices were calculated according to the original Bolton formula [12]. The mesio-distal widths obtained manually with the digital calliper and the Bolton Index calculated from these measurements were compared with those provided by the Bolton analysis tool available in the Invisalign Clin-Check Pro 6.0 software. 

To assess if the software uses the same formula proposed by Bolton in 1962, the Bolton discrepancy delivered by the software was compared with the one calculated with the Bolton formula on the basis of the software tooth width measurements.

To assess the agreement between methods (manual method and Clin-Check Pro 6.0 software), the tooth width measurements obtained manually on 54 the plaster casts, were compared with the ones delivered by the software on 54 digital casts obtained by the same patients. A total of 1296 repeated measurements were compared. 

To assess the software reliability (within method reliability), a pool of 402 clin-checks was selected. The pool consisted of 201 pairs of clin-checks corresponding to two consecutive treatment phases (T^1^ and T^2^) of the same patient with no stripping performed in between. The mesio-distal tooth widths as reported in the Bolton analysis tool at T^1^ and T^2^ were recorded, as well as the magnitude of the movement planned in the T^1^ clin-check in terms of translation, rotation, and angulation at the tooth crown level.

Measurements of mesiodistal width at T^1^ and T^2^ were compared to evaluate the reliability of the software. The Bolton anterior and overall ratio were compared at T^1^ and T^2^ to assess the agreement between the two values. The degree of agreement between the values at T^1^ and T^2^ was then correlated with the magnitude of the movement planned in the T^1^ clin-check in terms of translation, rotation, and angulation at the tooth crown level to assess if the degree of misalignment can affect the accuracy of the measurements.

### Statistical Analysis

To assess the inter- and intra-operator reliability and the Clin-Check Pro 6.0 software reliability, the Dahlberg’s formula, the coefficient of variability (CV), and the intraclass correlation coefficient (ICC) were calculated. To evaluate the agreement between the manual measurements obtained by the digital calliper tool and those delivered by the Clin-Check Pro 6.0 software, a paired *t*-test was used after assessing the normal distribution of the sample through a Kolmogorov–Smirnov test. Agreement was further studied by a regression model.

The same model was used to explore the agreement between the repeated measurements delivered by the Clin-Check Pro 6.0 software. A multiple linear regression model was applied to assess the effect of the planned movement on software reliability. Statistical analyses were performed with IBM SPSS Statistics software version 23.0 (IBM Corp., Armonk, NY, USA) at a 5% significance level (α = 0.05).

## 3. Results

### 3.1. Intra- and Inter-Operator Reliability

As reported in Table 1, regarding intra-operator reliability, the mean difference between repeated measurements was 0.01 ± 0.14 (SD) mm, being the difference not statistically significant (*p* = 0.199). The technical error of measurement (systematic and random) obtained by the Dahlberg’s formula was lower than 0.16 mm with a coefficient of variability lower than 1.71% and an ICC always higher than 0.9%. All reported parameters highlight the high intra-operator reliability. According to inter-operator reliability at a global level, the mean difference between repeated measurements was 0.01 ± 0.21 (SD) mm, the difference being not statistically significant (*p* = 0.110). The Dahlberg error was lower than 0.26 mm with a coefficient of variability lower than 2.63% and an ICC always higher than 0.80%. The mean width of 1.6 and 4.6 presented a statistically significant difference between the two operators, but all differences were lower than 0.5 mm.

### 3.2. Clin-Check Pro 6.0 Software within Method Reliability (Repeatablility)

As highlighted in Table 2, the repeated measurements performed by Clin-Check Pro 6.0 software displayed a random bias close to zero (−0.02 mm ≤ Δ ≤ 0.02 mm) for most measurements (91.7%). Only for tooth 1.6, the difference between the mean width at T^1^ was statistically different from the one at T^2^ (Δ = 0.04 ± 0.22 mm, *p* = 0.019). The Dahlberg error was lower than 0.10 mm for 75% of the measurements, and lower than 0.15 for 87.5% of them. The ICC was higher than 0.90 for most tooth widths (91.7%) and always greater than 0.80. The CV was lower than 1.5% in 91.7% of the cases. At a global level (Table 2), the ICC was equal to 0.99 and the CV was equal to 1.31%. According to the overall and anterior Bolton Indices, the system also displayed a low random error (−0.09; −0.05 mm) being the difference between the first and the second measurement non-statistically significant (*p* = 0.084 and *p* = 0.173). The Dahlberg error was 0.53 and 0.34 mm, respectively, while the ICC was high in both cases (0.91; 0.93). 

A multiple linear regression model was applied to explore how the planned movement between T^1^ and T^2^, considered as an indirect indicator of misalignment, can affect the accuracy of the method. As highlighted in Table 3, some planned tooth movements between the two clin-checks had a statistically significant impact on the accuracy of the method, but a definite pattern could not be identified, and despite statistical significance, the effect was low. The low coefficient of determination (r^2^) displayed by the multiple linear regression model indicates that its predictive value was null.

### 3.3. Agreement between the Manual Method and Clin-Check Pro 6.0 Software 

The accuracy of the software was assessed through agreement with the manual measurements (gold standard). When the two methods were compared (Table 4), the differences between the mean of the measurements were statistically significant, except for tooth 3.1. All values were negative, highlighting how Clin-Check Pro 6.0 software tends to provide larger widths compared with the manual method. 

The agreement was lower for the molars, especially in the upper ones, which display, respectively, a mean difference of 0.60 mm and 0.65 mm, while the difference was 0.53 mm and 0.48 mm for the lower molars. In the upper molars, the CV was high (4.97% − 4.46%) and the ICC was low (0.24 ≤ ICC ≤ 0.37). If molars were excluded, the mean difference between methods was low and always less than 0.18 mm, the Dahlberg error of measurement was never greater than 0.23 mm and the CV was less than 3.29%. In 75% of the positions the ICC was equal to or greater than 0.80. At a global level, the mean difference between the methods was −0.19 mm, with d = 0.24 mm, CV = 3.15%, and ICC = 0.98. 

The overall and anterior Bolton Indices provided by the two methods had a moderate correlation (ICC = 0.59; 0.69). The difference was statistically significant with a moderate Dahlberg error (1.26; 0.64 mm). When a regression model was applied to the data (Figure 1, Table 5), it was noticed that the bias was not constant, but it was higher for more negative values, negative values being the ones indicating a mandibular tooth size excess. To deeper analyse this behaviour, we compared the Bolton values delivered by Clin-Check Pro 6.0 software with those that could be obtained through the original Bolton formula using the tooth size widths as delivered by the software. It was possible to highlight (Table 4) that despite the very high ICC in both cases, the differences were statistically significant. We are not aware of the algorithm used by the software for the Bolton calculation, as it is a proprietary system of Align Technology, but as highlighted by the regression model (Figure 2, Table 5), the bias was not constant, but increased for more negative values and lowered as the tooth size discrepancy values became closer to zero or became positive.

## 4. Discussion

This is the first article reporting on the reliability of odontometric measurements delivered by the Clin-Check Pro 6.0 software. The sample size used in the study was set using a power analysis and is larger compared with the one used by other authors reporting on the same topic using different software. Reuschl et al., Koretsi et al., Naidu and Freer, Quimby et al., Soto-Alvarez et al., Sousa et al., Grunheid et al., and Camardella et al., performed accuracy and reliability appraisals using a sample size ranging from 19 to 50 plaster and digital casts [10,13,14,15,16,17,18,19].

### 4.1. Inter- and Inter-Operator Reliability 

In odontometric studies when assessing intra- and inter-examiner reliability, there is a consensus to consider for most outcomes the cut-off values of 0.5 mm for the Dahlberg error and 0.75 for ICCs as acceptable [11,20]. Moreover earlier studies showed that when a single operator is repeating dental width measurements on plaster casts at different time points the error itself is about 0.2 mm in average, and can be considered a normal error related to the procedure [21]. 

In the present study when the inter-and intra-operator reliability were assessed by means of the ICC all values were well-beyond 0.75, being all higher than 90 and the Dahlberg error, and CV, all lower than the cut off value. As highlighted by Koretsi et al., 2019 the assessment of the Dahlberg error, as opposed to the correlation coefficients, is valuable in reliability studies since it provides a quantitative assessment of error [11]. 

Inter-and intra-operator reliability can be considered substantial and consistent with the reference values reported by other authors in the odontometric field [10].

### 4.2. Clin-Check Pro 6.0 Software within Method Reliability (Repeatablility)

Clin-Check Pro 6.0 software displays extremely high reliability, with a random error close to zero. Even if the ICC, CV, and Dahlberg error are similar, at a global level, to the ones displayed by the manual method (intra-operator reliability), we can state that the software is more reliable since the SD and confidence intervals are lower. Koretsi et al. (2018) reporting on the reliability of Ivoris^®^analyze3D (Computer Konkret) found an ICC ranging from 0.7 to 0.86 for digital repeated measurement, that is, lower than the 0.86 to 0.99 range of Clin-Check Pro 6.0 software. Moreover, Koretsi et al. (2018) did not include molars in their appraisal. According to other authors and in agreement with our results, both upper and lower molars are the teeth characterized by the highest measurement errors [22,23].

Clin-Check Pro 6.0 software displays an extremely high reliability, with a random error close to zero. Even if the ICC, CV, and Dahlberg error are similar, at a global level, to the ones displayed by the manual method (intra-operator reliability), we can state that the software is more reliable since the SD and confidence intervals are lower. Koretsi et al. reporting on the reliability of Ivoris^®^analyze3D (Computer Konkret) found an ICC ranging from 0.7 to 0.86 for digital repeated measurement, that is to say, lower than the 0.86 to 0.99 range of the Clin-Check Pro 6.0 software. Moreover, Koretsi et al. did not include molars in their appraisal. According to other authors and in agreement with our results, both upper and lower molars are the teeth characterized by the highest measurement errors [22,23]. Reuschl et al. (2016) assessing the reliability of Orthoanalyzer software, found that while the digital method had a high accuracy in the frontal and premolar region, the error in the molar region was higher especially in the most distal teeth [13].

### 4.3. Agreement between the Manual Method and Clin-Check Pro 6.0 Software 

In the first phase, the study compares tooth width as measured by a digital calliper on the plaster casts and the ones delivered by the software.

Previous reports support the high level of agreement of dental measurements performed on plaster and digital casts [24]. Some authors reported an extremely high agreement with no statistically significant difference between the two methods regarding tooth width measurements, as is the case of Soto-Alvarez et al. who reported a mean difference of −0.007 mm to −0.136 mm and Rajshekar who found differences as small as 0.004–0.062 mm [16,24]. Most of the authors, on the contrary, in agreement with our results, found differences that were statistically significant but in all cases were considered clinically negligible [6]. Santoro et al. compared the accuracy of OrthoCAD (Cadent, Fairview, NJ, USA) and plaster models for tooth size measurements and reported statistically significant differences ranging from −0.16 to −0.38 mm [21]. Bootvong et al. also found statistically significant mean differences for virtual and plaster casts but all lower than 0.3 mm [25]. Considering that according to what was previously reported, the mean error in single observer repeated measurements is generally reported to be about 0.2 mm [3,22], the authors deemed these differences not significant from a clinical standpoint. Many authors in the odontometric field support that differences in tooth width measurements lower than 0.5 mm are not clinically significant, the differences mainly due to intra or interobserver random error can be positive or negative and due to their random nature the net effect on a pool of measurements can be negligible [3,26].

According to our results, the Clin-Check Pro 6.0 software consistently provides larger widths compared with the manual method. This finding was also reported by other authors reporting on digital measurements and can be due to the difficulty for calliper tips to access the contact point due to the bulk of the tips [10]. Moreover, in case of crowding, the contact point cannot be fully accessible. Digital methods usually allow image tools to be used that, by magnifying or rotating, facilitate access to the contact point in case of severe crowding [11]. We are not aware of how the software operates since it is a proprietary system of Align Technology, but the systematic bias displayed compared with the manual method can be partly due to a difference in the definition of mesiodistal width. In our study, it is defined as the greatest distance between physiological contact points, whereas the software in its algorithm probably gathers the maximum width along a mesiodistal axis. This can explain why a suboptimal agreement between methods is displayed, especially on upper molars. Probably due to their trapezoidal shape, there can be a higher mismatch between the line of maximum width and the line joining the contact points. On the contrary, lower incisors have a more regular shape, and an almost perfect match between the greatest distance between physiological contact points and the maximum width. These features can facilitate the location of the contact points and partially explain the higher agreement found in these teeth between the two methods.

Despite the statistically significant differences in tooth width measurements compared with the manual method, the Clin-Check Pro 6.0 software displayed an acceptable accuracy on all teeth except for the upper molars since the ICC (0.75), Dahlberg error (0.5 mm), and CV (5%) cut-off value were not outreached. If molars are excluded, only 12 out of 1080 differences in tooth width exceed 0.5 mm, equivalent to 1.11% of the data, in agreement with what was reported by Naidu and Freer (2013) on the iOC/OrthoCAD system (Cadent, Fairview, NJ, USA) [14].

According to the results of our study, the difference between the Bolton Index values delivered by the software and the one calculated with the original Bolton formula present differences higher than 2 mm, especially when maxillary or mandibular excess display a high value. We could not find any study investigating the extent of Bolton discrepancy, which can eventually influence treatment planning decision-making. However, in the orthodontic literature, the clinically relevant thresholds for mean differences of linear measurements based on more than two landmarks are set at 2.0 mm [27,28]. Differences of up to two millimetres are considered acceptable, as they are unlikely to cause a change in the treatment plan [10]. Once again we should underline that we are not aware of the algorithm used by the software for Bolton calculation, but as highlighted by the regression model the bias can be higher than 2 mm especially in case of a mandibular tooth size excess, suggesting to be cautious in relying on the software data for decision-making. One of the limitations of the current study relies on the closed nature of the Invisalign system, which did not allow for the same STL to be measured twice, thus forcing the repeatability measurements to be performed on two different STL related to different treatment stages. The introduced bias can however be considered low since according to the exclusion criteria stripping of the interproximal surfaced should not be performed in between the two treatment phases and the STL were gathered by the same intra-oral scanning device.

## 5. Conclusions

Tooth width measurements delivered by the Clin-Check Pro 6.0 software, despite the positive bias, can be considered accurate and clinically acceptable for all teeth except molars. Planned tooth movements as an indirect indicator of tooth misalignment seem not to have an association with the system reliability.

The software does not use the original Bolton formula for tooth discrepancy assessment and the anterior and overall Bolton made available by the software is not accurate and clinically acceptable especially in case of mandibular excess.

The availability of reliable odontometric data makes Clin-Check Pro 6.0 software a source of tooth measurements with possible implications in the field of odontometric research.

## Figures and Tables

**Figure 1 ijerph-19-03581-f001:**
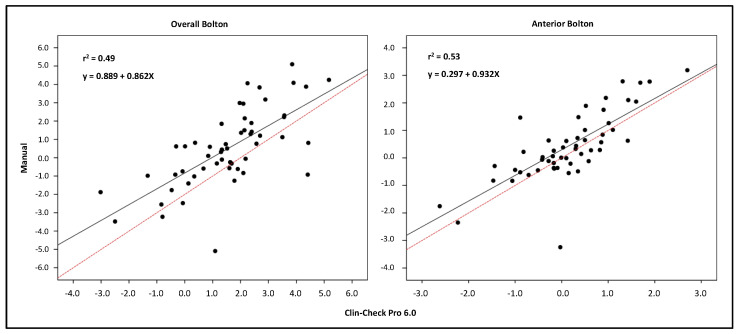
Linear regression analysis of the Bolton Index (overall and anterior) showing the correlation between the manual method and the Clin-Check Pro 6.0 software. The equation, the coefficient of determination (r^2^), the regression lines (solid), and the line of perfect correlation (dashed) are shown in each graph.

**Figure 2 ijerph-19-03581-f002:**
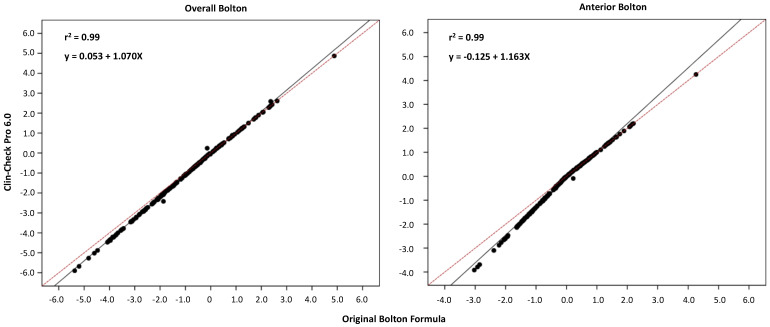
Linear regression analysis of the Bolton Index (overall and anterior) showing the association between the Clin-Check Pro 6.0 software values and the ones obtained through the original Bolton formula. The equation, the coefficient of determination (r^2^), the regression line (solid), and the line of perfect correlation (dashed) are shown in each graph.

**Table 1 ijerph-19-03581-t001:** Intra and inter-operator reliability and error within tooth width measurements on plaster casts.

	Intra-Examiner Reliability						Inter-Examiner Reliability					
Tooth (FDI)	Δ (SD)	95% CI	*p*	Dahlberg Error (mm)	CV (%)	ICC	Δ (SD)	95% CI	*p*	Dahlberg Error (mm)	CV (%)	ICC
1.1	−0.03 (−0.09)	−0.09/0.04	0.336	0.06	0.76	0.98	−0.01 (0.14)	−0.12/0.09	0.798	0.1	1.13	0.96
1.2	0.01 (−0.14)	−0.09/0.10	0.912	0.09	1.4	0.98	0.11 (0.25)	−0.08/0.29	0.211	0.18	2.75	0.92
1.3	0.06 (0.14)	−0.04/0.16	0.218	0.1	1.35	0.93	0.07 (0.24)	−0.1/0.24	0.39	0.17	2.22	0.80
1.4	0.01 (0.12)	−0.07/0.09	0.811	0.08	1.11	0.96	0.01 (0.12)	−0.08/0.09	0.894	0.08	1.1	0.96
1.5	−0.03 (0.14)	−0.13/0.07	0.553	0.1	1.46	0.97	−0.04 (0.15)	0.15/0.07	0.422	0.1	1.55	0.97
1.6	0.11 (0.19)	−0.03/0.24	0.103	0.14	1.39	0.93	0.24 (0.3)	0.03/0.45	**0.031 ***	0.26	2.5	0.83
2.1	−0.04 (0.13)	−0.13/0.05	0.385	0.09	1.06	0.97	−0.07 (0.22)	−0.23/0.09	0.343	0.16	1.83	0.94
2.2	0.01 (0.13)	−0.08/0.10	0.776	0.09	1.3	0.97	0.07 (0.28)	−0.13/0.27	0.455	0.19	2.89	0.88
2.3	−0.03 (0.13)	−0.12/0.06	0.449	0.09	1.15	0.95	0.02 (0.22)	−0.14/0.18	0.777	0.15	1.9	0.83
2.4	0.02 (0.12)	−0.07/0.10	0.667	0.08	1.16	0.97	−0.01 (0.21)	−0.15/0.14	0.917	0.14	1.95	0.91
2.5	−0.03 (0.14)	−0.13/0.07	0.491	0.1	1.44	0.98	−0.04 (0.21)	−0.19/0.01	0.52	0.14	2.12	0.94
2.6	0.04 (0.14)	−0.06/0.14	0.374	0.1	0.97	0.96	0.04 (0.26)	−0.15/0.23	0.667	0.18	1.73	0.88
3.1	−0.03 (0.14)	−0.12/0.07	0.547	0.09	1.75	0.93	−0.03 (0.15)	−0.14/0.08	0.527	0.1	1.92	0.91
3.2	0.02 (0.11)	−0.06/0.10	0.556	0.07	1.27	0.96	−0.02 (0.14)	−0.12/0.09	0.73	0.1	1.64	0.94
3.3	−0.06 (0.11)	−0.14/0.02	0.103	0.09	1.29	0.96	−0.06 (0.14)	−0.15/0.02	0.118	0.09	1.34	0.96
3.4	−0.05 (0.14)	−0.15/0.05	0.289	0.1	1.4	0.91	−0.05 (0.14)	−0.15/0.05	0.268	0.1	1.4	0.91
3.5	0.06 (0.13)	−0.03/0.15	0.188	0.1	1.35	0.98	−0.04 (0.45)	−0.36/0.28	0.788	0.3	4.17	0.80
3.6	0.11 (0.20)	−0.03/0.25	0.121	0.16	1.42	0.95	0.15 (0.25)	−0.04/0.32	0.095	0.19	1.78	0.92
4.1	0.01 (0.09)	−0.05/0.07	0.755	0.06	1.14	0.95	0.02 (0.11)	−0.06/0.09	0.635	0.07	1.42	0.92
4.2	0.04 (0.14)	−0.06/0.14	0.384	0.1	1.71	0.94	0.03 (0.21)	−0.12/0.18	0.692	0.14	2.43	0.89
4.3	−0.02 (0.13)	−0.11/0.08	0.702	0.09	1.29	0.95	−0.04 (0.26)	−0.23/0.15	0.661	0.18	2.63	0.82
4.4	0.04 (0.13)	−0.05/0.14	0.353	0.09	1.32	0.96	0.05 (0.17)	−0.07/0.18	0.354	0.12	1.71	0.93
4.5	0 (0.16)	−0.12/0.11	0.938	0.11	1.48	0.96	−0.06 (0.22)	−0.21/0.10	0.432	0.15	2.07	0.93
4.6	0.06 (0.10)	−0.01/0.13	0.095	0.08	0.72	0.98	0.11 (0.15)	0.01/0.21	**0.040 ***	0.12	1.13	0.96
**Total**	0.01 (0.14)	−0.01/0.03	0.199	0.1	1.28	0.99	0.01 (0.21)	0.13 0.08	0.110	0.15	1.97	0.99

Δ: Mean difference between repeated measurements; FDI: Fédération Dentaire Internationale notation; SD: standard deviation; CI: confidence interval; CV: coefficient of variability; ICC: intraclass correlation coefficient; *t*-test * *p* < 0.05.

**Table 2 ijerph-19-03581-t002:** Clin-Check Pro 6.0 software, within method reliability.

Tooth (FDI)	Δ (SD)	95% CI	*p*	Dahlberg Error (mm)	CV (%)	ICC
1.1	0.00 (0.11)	−0.01/0.02	0.767	0.08	0.86	0.98
1.2	0.00 (0.09)	−0.01/0.02	0.642	0.07	0.97	0.98
1.3	−0.01 (0.11)	−0.02/0.01	0.335	0.08	0.99	0.97
1.4	0.01 (0.12)	−0.01/0.03	0.280	0.09	1.20	0.96
1.5	0.01 (0.11)	−0.01/0.02	0.280	0.08	1.17	0.96
1.6	0.04 (0.22)	0.01/0.07	**0.019 ***	0.16	1.46	0.93
2.1	0.00 (0.10)	−0.02/0.01	0.719	0.07	0.82	0.99
2.2	0.01 (0.12)	−0.01/0.02	0.491	0.08	1.20	0.97
2.3	0.01 (0.14)	−0.01/0.03	0.459	0.10	1.26	0.95
2.4	−0.01 (0.12)	−0.02/0.01	0.502	0.08	1.15	0.97
2.5	0.01 (0.13)	0.00/0.03	0.099	0.09	1.29	0.95
2.6	0.02 (0.21)	−0.01/0.05	0.211	0.15	1.39	0.93
3.1	−0.02 (0.23)	−0.05/0.01	0.208	0.16	3.10	0.83
3.2	−0.01 (0.12)	−0.03/0.01	0.255	0.08	1.39	0.96
3.3	−0.02 (0.13)	−0.03/0.00	0.098	0.09	1.35	0.96
3.4	0.00 (0.10)	−0.01/0.01	0.984	0.07	1.03	0.97
3.5	0.03 (0.23)	0.09/0.06	0.061	0.16	2.24	0.86
3.6	−0.01 (0.15)	−0.03/0.01	0.540	0.11	0.95	0.97
4.1	−0.01 (0.08)	−0.02/0.01	0.277	0.06	1.08	0.97
4.2	−0.01 (0.09)	−0.02/0.00	0.206	0.07	1.11	0.97
4.3	−0.01 (0.14)	−0.03/0.01	0.239	0.10	1.42	0.96
4.4	0.00 (0.10)	−0.02/0.01	0.751	0.07	1.01	0.97
4.5	0.01 (0.11)	−0.01/0.02	0.519	0.08	1.10	0.96
4.6	−0.01 (0.15)	−0.03/0.01	0.557	0.11	0.94	0.97
Total	0.002 (0.14)	−0.002/0.006	0.400	0.10	1.31	0.99
Overall Bolton	−0.09 (0.74)	−0.19/0.01	0.084	0.53	--	0.91
Anterior Bolton	−0.05 (0.48)	−0.11/0.02	0.173	0.34	--	0.93

Δ: Mean difference between repeated measurements; FDI: Fédération Dentaire Internationale notation; SD: Standard Deviation; CI: Confidence interval; CV: coefficient of variability; ICC: intraclass correlation coefficient; *t*-test * *p* < 0.05.

**Table 3 ijerph-19-03581-t003:** Correlation between planned movement and software reliability, only the teeth displaying a statistically significant correlation are reported.

	Translation (BL)		Rotation (MD)		Angulation (MD)		
Tooth (FDI)	ICC (95% CI)	*p*	ICC (95% CI)	*p*	ICC (95% CI)	*p*	r^2^
1.3	0.004 (−0.011/0.019)	0.59	0.002 (0.000/0.003)	**0.039 ***	0.002 (−0.003/0.007)	0.374	4.50%
1.4	0.001 (−0.018/0.020)	0.936	0.005 (0.001/0.008)	**0.007 ****	−0.001 (−0.006/0.004)	0.663	4.00%
1.5	0.011 (−0.001/0.023)	0.078	0.003 (0.000/0.005)	**0.030 ***	−0.002 (−0.006/0.002)	0.312	5.60%
1.6	−0.026 (−0.066/0.014)	0.204	0.011 (0.006/0.016)	**<0.001 *****	−0.003 (−0.014/0.009)	0.674	8.40%
2.1	0.001 (−0.008/0.010)	0.809	0.000 (−0.001/0.002)	0.642	0.008 (0.003/0.013)	**<0.001 *****	6.10%
2.3	0.035 (0.016/0.053)	**<0.001 *****	−0.001 (−0.003/0.001)	0.393	0.009 (0.003/0.015)	**0.005 ****	13.00%
3.4	0.017 (0.001/0.033)	**0.043 ***	−0.001 (−0.003/0.001)	0.164	0.002 (−0.002/0.005)	0.357	2.70%
3.5	0.056 (0.017/0.096)	**0.006 ****	0.002 (−0.002/0.006)	0.359	0.006 (−0.005/0.017)	0.254	6.90%
4.5	−0.004 (−0.021/0.012)	0.618	0.001 (−0.001/0.003)	0.509	0.004 (0.000/0.009)	**0.047 ***	2.40%
4.6	−0.021 (−0.050/0.008)	0.162	0.009 (0.003/0.015)	**0.005 ****	0.006 (−0.001/0.014)	0.109	5.30%

BL: buccolingual; MD: mesiodistal; FDI: Fédération Dentaire Internationale notation; ICC: intraclass correlation coefficient; CI: confidence interval; r^2^: coefficient of determination; * *p* < 0.05; ** *p* < 0.01; *** *p* < 0.001.

**Table 4 ijerph-19-03581-t004:** Agreement between manual method and Clin-Check Pro 6.0 software.

	Agreement between Methods					
Tooth (FDI)	Δ (SD)	95% CI	*p*	Dahlberg Error (mm)	CV (%)	ICC
1.1	−0.18 (0.28)	−0.25/−0.10	**<0.001 *****	0.24	2.74	0.86
1.2	−0.16 (0.23)	−0.22/−0.10	**<0.001 *****	0.20	2.91	0.90
1.3	−0.16 (0.20)	−0.22/−0.11	**<0.001 *****	0.18	2.31	0.85
1.4	−0.14 (0.19)	−0.19/−0.09	**<0.001 *****	0.17	2.33	0.89
1.5	−0.18 (0.20)	−0.24/−0.13	**<0.001 *****	0.19	2.74	0.89
1.6	−0.65 (0.39)	0.75/−0.54	**<0.001 *****	0.53	4.97	0.24
2.1	−0.16 (0.17)	−0.21/−0.11	**<0.001 *****	0.17	1.91	0.92
2.2	−0.18 (0.22)	−0.24/−0.10	**<0.001 *****	0.20	2.97	0.88
2.3	−0.14 (0.24)	−0.20/−0.07	**<0.001 *****	0.20	2.52	0.80
2.4	−0.11 (0.30)	−0.19/−0.03	**0.011 ***	0.22	3.10	0.80
2.5	−0.18 (0.18)	0.23/−0.13	**<0.001 *****	0.18	2.63	0.85
2.6	−0.60 (0.30)	−0.69/−0.52	**<0.001 *****	0.48	4.46	0.37
3.1	−0.02 (0.25)	−0.09/0.05	0.570	0.18	3.29	0.76
3.2	−0.14 (0.20)	−0.19/−0.09	**<0.001 *****	0.17	2.88	0.82
3.3	−0.16 (0.29)	−0.24/−0.08	**<0.001 *****	0.23	3.22	0.75
3.4	−0.11 (0.22)	−0.17/−0.05	**0.001 ****	0.17	2.39	0.84
3.5	−0.18 (0.29)	0.26/0.10	**<0.001 *****	0.24	3.26	0.78
3.6	−0.31 (0.26)	−0.38/−0.24	**<0.001 *****	0.29	2.58	0.82
4.1	−0.09 (0.15)	−0.13/−0.05	**<0.001 *****	0.12	2.24	0.87
4.2	−0.09 (0.25)	−0.16/0.02	**0.001 ****	0.18	3.09	0.78
4.3	−0.14 (0.25)	−0.21/−0.07	**<0.001 *****	0.20	2.97	0.79
4.4	−0.11 (0.21)	−0.16/−0.05	**<0.001 *****	0.17	2.33	0.85
4.5	−0.14 (0.27)	−0.21/−0.06	**<0.001 *****	0.21	2.86	0.82
4.6	−0.19 (0.27)	−0.26/−0.12	**<0.001 *****	0.23	2.08	0.87
Total	−0.19 (0.28)	−0.20/−0.17	**<0.001 *****	0.24	3.15	0.98
Overall Bolton	0.95 (1.93)	0.53/1.37	**<0.001 *****	1.26	--	0.59
Anterior Bolton	0.29 (0.86)	0.05/0.52	**0.018 ***	0.64	--	0.93

Δ: Mean difference between methods; FDI: Fédération Dentaire Internationale notation; SD: standard deviation; CI: confidence interval; CV: coefficient of variability; ICC: intraclass correlation coefficient; *t*-test * *p* < 0.05; ** *p* < 0.01; *** *p* < 0.001.

**Table 5 ijerph-19-03581-t005:** Slope and constant at origin of the regression analyses, for overall and anterior Bolton between the Clin-Check Pro 6.0 software values and the original Bolton formula values.

Clin-Check Pro 6.0 Software/Original Bolton Formula	r^2^	Slope [CI 95%]	Constant [CI 95%]
Overall Bolton	0.99	1.070 [1.065, 1.075]	−0.053 [−0.063, −0.043]
Anterior Bolton	0.99	1.163 [1.150, 1.176]	−0.125 [−0.140, −0.111]

CI: 95% confidence intervals; r^2^: determination coefficient.

## Data Availability

All data generated or analysed during this study are included in this published article.

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
