# Peer review of "Reliability of Tooth Width Measurements Delivered by the Clin-Check Pro 6.0 Software on Digital Casts: A Cross-Sectional Study"

_ijerph, 2022, doi:10.3390/ijerph19063581_

Round 1

Reviewer 1 Report

This article compares stability and accuracy of teeth width measurements between human operator and automated diagnostic software (Clin-Check pro). Manual measurements were made on standard plaster cast by experienced operator using digital calliper. Automated measurements were performed using digital intraoral scanner and corresponding software. Bolton indexes (one of standard orthodontics) were also calculated. Both methods proven to deliver constant and stable results. But agreement of both methods was not complete. Dimensions obtained for lower molars significantly differed. Also, Boltons indexes calculated by the software were not suitable for clinical practice. Conclusion of the article is that automated software measurements of teeth dimensions is highly effective and could be used mainly for research purposes. 

Strengths of the article: 

The question of accuracy and repeatability of automated measurements is highly actual 

Methodology of the research is good prepared and clearly described 

Statistical analysis and results are clear and well presented 

This article could help many researchers and clinical orthodontist in their daily practise 

Weaknesses of the article: 

It is not clear how and why Boltons indexes calculated by the software differs from manually calculated ones 

Recommendation 

Please explain why Bolton indexes calculated by the software are not suitable for clinical use 

Author Response

Dear Referee 1 thank you so much for the time and effort spent in reviewing our article

It is not clear how and why Boltons indexes calculated by the software differs from manually calculated ones: 

(Line 230-234) We are not aware of the algorithm used by the software for Bolton calculation since it is a proprietary system of Align Technology, but as highlighted by the regression model (Figure 2), the bias was not constant but higher for more negative values and lower as the tooth size discrepancy values got closer to zero or became positive.

Please explain why Bolton indexes calculated by the software are not suitable for clinical use.

According to the results of our study the difference between the Bolton index delivered by the software and the one calculated with the original Bolton formula present differences higher than 2 mm, especially when maxillary or mandibular excess display a high value. Many authors supports that differences in the overall or anterior Bolton calculation, up to two millimeter can be considered clinically acceptable since they unlikely cause a change in the treatment plan.

A detailed explanation have been added according to yours suggestion (line 343-355)

Reviewer 2 Report

Title:
- too long, should be shortened and modified
- the authors should consider whether they really want to include the software name in the title.
- Or should the paper be an advertisement for the software? If it is contract research, this should be stated in case of conflict of interest.

Abstract:
"extremely time efficient" should be modified

Introduction:
- Invisalign should be mentioned without ®-sign and correctly with company, place and country, also the corresponding software.
- "Moreover, we wanted to assess the agreement" should be modified
- For the department of uni Valencia the country "Spain" should be specified

Materials and Methods:
- Why was each tooth measured only twice and averaged?
- What does "one trained operator" mean? What kind of calibration was performed? This should be described in detail. Was only one operator calibrated or several? The reliability of the operator should be described in more detail and not just quoted by reference.
- The abstract says "trained and calibrated operator", this calibration is not adequately described.

Discussion:
- The discussion needs a deep revision, both in content and language

- thematic sections would be desirable.
- At the beginning of the discussion, the goal of the study should first be presented with the main results. A "common thread" should run through each section of the discussion to generate a flow of thought.
- OrthoCAD® as stated correctly above with company, location and country.

Conclusions:
- Conclusions should be revised to only accurately represent the conclusion obtained through the study.

Author Response

Dear Referee 2 thank you so much for the time and effort spent in reviewing our article. We answer here to you comments that are also highlighted in the main text.

Title:

- too long, should be shortened and modified

Dear Referee 2, according to your suggestion the title have been shortened, but the study design was added according to Referee 3 suggestion.

- the authors should consider whether they really want to include the software name in the title.

- Or should the paper be an advertisement for the software? If it is contract research, this should be stated in case of conflict of interest.

Dear Referee 2, as stated in the conflict of interest declaration, we have no financial/personal interest or belief that could affect our objectivity and we did not received any funding from Align Technology. The article is not intended to be an advertisement of the software, that have no commercial relevance per se since is can be accessed for free in the online platform provided by Invisalign to design the treatment plan. We do believe that it is important to maintain the name of the software in the title, since the article wants to be a guide in handling and interpretate the data delivered by this specific software. But If you still consider that we should remove it from the title we will do it.

Abstract:
"extremely time efficient" should be modified

Dear Referee 2 we removed the statement about time efficiency since as you highlighted since it is not a conclusion that have been obtained through the study results.

Introduction:
- Invisalign should be mentioned without ®-sign and correctly with company, place and country, also the corresponding software.

Dear Referee 2, the ®-sign have been removed according to your suggestion and the company, place and country have been indicated when Invisalign and the software are mentioned for the first time.

- "Moreover, we wanted to assess the agreement" should be modified

Dear Referee 2, the section have been rephrased according to your suggestion.

- For the department of uni Valencia the country "Spain" should be specified

Dear Referee 2, country have been specified as indicated.

Materials and Methods:
- Why was each tooth measured only twice and averaged?

Dear Referee 2, as indicated in the text and by the relative citation [9] we used the tooth width measuring method as described by Flores-Mir et al. in 2003. Anyway, the protocol of measuring twice and averaging if the difference is lower than 0.2 mm was previously reported by many authors and can be considered a widely accepted protocol in odontometric studies. I enclose a few citation of previous studies reporting the same protocol.

Santoro M, Ayoub ME, Pardi VA, Cangialosi TJ. Mesiodistal crown dimensions and tooth size discrepancy of the permanent dentition of Dominican Americans. Angle Orthod. 2000 Aug;70(4):303-7. doi: 10.1043/0003-3219(2000)070<0303:MCDATS>2.0.CO;2. PMID: 10961780.

Richardson ER, Malhotra SK. Mesiodistal crown dimension of the permanent dentition of American Negroes. Am J Orthod. 1975 Aug;68(2):157-64. doi: 10.1016/0002-9416(75)90204-3. PMID: 1056704.

Bishara SE, Jakobsen JR, Abdallah EM, Fernandez Garcia A. Comparisons of mesiodistal and buccolingual crown dimensions of the permanent teeth in three populations from Egypt, Mexico, and the United States. Am J Orthod Dentofacial Orthop. 1989 Nov;96(5):416-22. doi: 10.1016/0889-5406(89)90326-0. PMID: 2816841.

Bernabé E, Villanueva KM, Flores-Mir C. Tooth width ratios in crowded and noncrowded dentitions. Angle Orthod. 2004 Dec;74(6):765-8. doi: 10.1043/0003-3219(2004)074<0765:TWRICA>2.0.CO;2. PMID: 15673138.

- What does "one trained operator" mean? What kind of calibration was performed? This should be described in detail. Was only one operator calibrated or several? The reliability of the operator should be described in more detail and not just quoted by reference.

According to your valuable suggestion the following text have been added:

The operator was trained by one senior team member (D.G.) through theoretical and practical sessions in which the use of the caliper and the landmark positioning and location were explained and performed, every training session involved the repeated measurement of three study cast, not belonging to the study group. The training session ended when the operator reached a main difference between repeated measurements, less or equal to 0.2 mm. (line 107-112)

The reliability of the second operator was assessed in a previous odontometric study[1], and the related reliability data are provided in supplemental table 1. (line 118)

- The abstract says "trained and calibrated operator", this calibration is not adequately described.

Dear Referee 2, find our answer in the previous point

Discussion:

- The discussion needs a deep revision, both in content and language

- thematic sections would be desirable.
- At the beginning of the discussion, the goal of the study should first be presented with the main results. A "common thread" should run through each section of the discussion to generate a flow of thought.

Dear Referee 2 the Discussion section have been totally rearranged according to your suggestions. If you still feel that a language revision is needed we will send the manuscript to the language author service of MDPI.

- OrthoCAD® as stated correctly above with company, location and country.

Corrected as indicated

Conclusions:
- Conclusions should be revised to only accurately represent the conclusion obtained through the study.

Conclusion have been rearranged according to your suggestions and the part about time efficiency that was not supported by the results have been removed

Reviewer 3 Report

Overall the presentation is in detail and logically written. However I have some comments.

The introduction is mostly background and history telling of orthodontic measurements and how much data the Clin-Check Pro potentially has. This section should introduce the topic, which narrows down to the research question and leading to the aim of the study, which is related to using digital measurement method in place of manual measurement. This would make the writing more focused. 

p.3 line 116, 117 typographic error

Materials & methods

  • It is unclear whether the scans were obtained intra-orally or by scanning plaster casts (or even scanning the silicon impression), although it was mentioned plaster casts were taken on the same session as the scan
  • ‚within method agreement‘ do you mean ‚repeatability‘?

p.3 line 136 do you mean paired t-test?

p.3 line 147 what is the value of 0.14? please specify whether it is a standard deviation, standard error, or something else. Same for p.4 line 152

p.4 line 153 statistical significance should be shown with a p-value

Line 154, 155 the unit is missing

Comment on tables: please always check consistency of the display, also please note Table 2 header row and second last row.

There are no p values less than 0.01 or 0.001 in tables 1 & 2 so the key to the table should not show something like **p<0.01.

Also please explain the direction of mean difference - does positive value mean manual or digital measurements are higher?

Any explanation for higher discrepancy for tooth 16 and higher agreement for tooth 31?

Page 6 line 182 please clarify the movement is due to scanning (as in causing an artefact) or is referring tooth movement (change in position) between time points

Page 6 line 184/185: check English sentence

In the discussion section there is a lengthy discussion on time efficiency of using digital measurements. However, this was not investigated or described in the materials and methods, and is not the aim of the study. Please consider reporting related time measurements, or sharpen the focus of the discussion to geometric measurements.

I also suggest to add the clinical implications of doing this study, as the discussion has mentioned, there is already much evidence supporting the use of digital casts and softwares for measurement because of their accuracy and reliability and differences are clinically insignificant (<0.5 mm). What is the goal of conducting this study by repeating the same methods? Please specify.

Author Response

Dear Referee 3 Thank you so much for the time and effort spent in reviewing our paper, all your remarks were highly appreciated. The required modification are reported here and highlighted in the main text.

The introduction is mostly background and history telling of orthodontic measurements and how much data the Clin-Check Pro potentially has. This section should introduce the topic, which narrows down to the research question and leading to the aim of the study, which is related to using digital measurement method in place of manual measurement. This would make the writing more focused. 

Dear Referee 3 the introduction have been modified according to your suggestion, I hope we have met your requirements.

p.3 line 116, 117 typographic error

The error was corrected as indicated

Materials & methods

  • It is unclear whether the scans were obtained intra-orally or by scanning plaster casts (or even scanning the silicon impression), although it was mentioned plaster casts were taken on the same session as the scan

Dear Referee 3 the scanner were obtained intra-orally to disambiguate the following statement have been added in the text:

Plaster casts were obtained from upper and lower impressions taken on the same session as the intra-oral scan (line 92-93)

  • ‚within method agreement‘ do you mean ‚repeatability‘?

Dear Referee 3, according to your suggestion the term repeatability have been added in the text at sub-paragraphs headings line 174 and 269

p.3 line 136 do you mean paired t-test?

Corrected as indicated

p.3 line 147 what is the value of 0.14? please specify whether it is a standard deviation, standard error, or something else. Same for p.4 line 152

Corrected as indicated

p.4 line 153 statistical significance should be shown with a p-value

Added as indicated

Line 154, 155 the unit is missing

1.6 and 4.6 are the FDI codes for upper right first molar and lower right first molar

Comment on tables: please always check consistency of the display, also please note Table 2 header row and second last row.

Table have been reviewed and the error in table 2 have been corrected as indicated

There are no p values less than 0.01 or 0.001 in tables 1 & 2 so the key to the table should not show something like **p<0.01.

Removed as indicated

Also please explain the direction of mean difference - does positive value mean manual or digital measurements are higher?

As indicated the following statement have been added (line 206-207):

All values were negative highlighting how the Clin-Check Pro 6.0 tends to provide larger widths when compared to the manual method.

Any explanation for higher discrepancy for tooth 16 and higher agreement for tooth 31?

According to your valuable suggestions the following have been added:

In our study it is defined as the greatest distance between physiological contact points, while the software in its algorithm probably gathers the maximum width along a mesiodistal axis. This could explain why a suboptimal agreement between methods is displayed, especially on upper molars. Probably due to their trapezoidal shape, there can be a higher mismatch between the line of maximum width and the line joining the contact points. On the contrary lower incisors have a more regular shape, and a almost perfect match between the greatest distance between physiological contact points and the maximum width. These features could facilitate contact point location and partly explain the higher agreement found in these teeth between the two methods.

Page 6 line 182 please clarify the movement is due to scanning (as in causing an artefact) or is referring tooth movement (change in position) between time points

According to your indications the following have been added:

As highlighted in Table 3, some planned tooth movements in between of the two clin-checks (line 194-195)

Page 6 line 184/185: check English sentence

Line was rephrased as indicated

In the discussion section there is a lengthy discussion on time efficiency of using digital measurements. However, this was not investigated or described in the materials and methods, and is not the aim of the study. Please consider reporting related time measurements, or sharpen the focus of the discussion to geometric measurements.

Dear Referee 3 according to Conclusion have been rearranged according to your suggestions and the part about time efficiency that was not supported by the results have been removed 

I also suggest to add the clinical implications of doing this study, as the discussion has mentioned, there is already much evidence supporting the use of digital casts and softwares for measurement because of their accuracy and reliability and differences are clinically insignificant (<0.5 mm). What is the goal of conducting this study by repeating the same methods? Please specify.

According to your valuable suggestions the following have been added:

Due to the fast development of digital technologies , a large variety of hardware and software solutions are continuously made available to the users and is therefore important to test their accuracy and reliability prior to use their finding for decision making in a clinical setting clinical setting. (line 56-59)

Moreover to justify the study you should take into account that, this is the first article reporting on the reliability of odontometric measurements delivered by the Clin-Check Pro 6.0 software.

Reviewer 4 Report

The Authors must see my remarks

Author Response

Dear Referee 4 Thank you so much for the time and effort spent in reviewing our paper, all your remarks were highly appreciated. The required modification are reported in the pdf. Where we tried to address all your suggestions. Moreover they are highlighted in the main text.

Round 2

Reviewer 2 Report

All points have been sufficiently addressed.